# Saliency-enhanced infrared and visible image fusion via sub-window variance filter and weighted least squares optimization

**Peicheng Wang[1], Tingsong Li[2], Pengfei Li [3]***

**1** School of Statistics and Mathematics, Guangdong University of Finance and Economics, Guangzhou, China, **2** School of Automation, Xi'an JiaoTong University, Xi'an, China, **3** School of SWJTU-LEEDS joint school, Southwest Jiaotong University, Chengdu, China

* xs2fpz@163.com

## Abstract

This paper proposes a novel method for infrared and visible image fusion (IVIF) to address the limitations of existing techniques in enhancing salient features and improving visual clarity. The method employs a sub-window variance filter (SVF) based decomposition technique to separate salient features and texture details into distinct band layers. A saliency map measurement scheme based on weighted least squares optimization (WLSO) is then designed to compute weight maps, enhancing the visibility of important features. Finally, pixel-level summation is used for feature map reconstruction, producing high-quality fused images. Experiments on three public datasets demonstrate that our method outperforms nine state-of-the-art fusion techniques in both qualitative and quantitative evaluations, particularly in salient target highlighting and texture detail preservation. Unlike deep learning-based approaches, our method does not require large-scale training datasets, reducing dependence on ground truth and avoiding fused image distortion. Limitations include potential challenges in handling highly complex scenes, which will be addressed in future work by exploring adaptive parameter optimization and integration with deep learning frameworks.

## 1. Introduction

High-quality image data is crucial for advancing detection technology and artificial intelligence in computer vision. However, limited by differences in imaging mechanisms, a single sensor often fails to capture the complete information of a scene [1]. For example, the imaging principle of infrared sensors is thermal radiation imaging, and the captured images tend to highlight thermal targets such as humans and vehicles, which are less affected by changes in light. Nevertheless, the drawbacks are more noise in the image, blurring of texture details and poor visualization. In contrast, the imaging principle of visible sensors is reflected light imaging, which captures

**Data availability statement:** All datasets used in this study are publicly available. The data can be accessed from the following repositories: 1, TNO – Available from TNO image fusion dataset at https://figshare.com/articles/TNO_Image_Fusion_Dataset/1008029 . 2, RoadScene – Available from "U2fusion: a unified unsupervised image fusion network" at DOI: 10.1109/TPAMI.2020.3012548. 3, LLVIP – Available from "LLVIP: A Visible-infrared Paired Dataset for Low-light Vision" at DOI: https://doi.org/10.1109/ICCVW54120.2021.00389.

**Funding:** The author(s) received no specific funding for this work.

**Competing interests:** The authors have declared that no competing interests exist.

images with a textured appearance and rich background information. However, it tends to be susceptible to light variations and suffers from severe information loss. It is necessary to effectively integrate images containing different modal feature information to generate a fused image with clear vision, prominent features and rich information. Therefore, infrared and visible image fusion (IVIF) techniques have emerged [2–6]. These techniques aim to enhance perceptual image quality by combining the strengths of both modalities, which is closely related to the field of perceptual image quality assessment [7–9]. In addition, tasks such as multi-exposure image fusion [10,11], multi-focus image fusion [12,13], multi-modal medical image fusion [14,15], and multi-spectral image fusion [16,17] have been derived based on the differences in information present in the images captured by the sensors. Moreover, High-quality images generated by image fusion techniques are widely employed in various fields, including object detection [18], target tracking [19], and remote sensing detection [20,21]. More importantly, recent advancements in multimodal fusion and visual saliency have demonstrated the importance of integrating complementary information and perceptual quality enhancement [22–24]. These studies provide valuable insights into the role of saliency-driven approaches and multimodal data integration, which inspire our work to improve the quality and interpretability of infrared and visible image fusion.

With the rapid development and iterative updating of filter theory and deep learning techniques, the exploration on image fusion techniques has become a research hotspot. At present, image fusion algorithms can generally be classified into two broad categories according to the type of algorithm, i.e., traditional image fusion methods based on mathematical theory and fusion methods based on deep learning. Among them, the traditional image fusion algorithms primarily include methods based on multi-scale transform (MST) [3,25], subspace-based [26] and sparse representation (SR) -based [27]. Such methods usually do not require heavy and complex training and often rely on manually designed fusion strategies. For example, MST-based methods often perform image decomposition through some kind of filter, such as weighted least squares filter [28], cross-bilateral filter [29], etc., and then fusion and reconstruction of the multiscale layers are performed by hand-designed fusion strategies. Subspace-based techniques frequently map characteristics of high-dimensional data into lower-dimensional spaces by some kind of dimensionality reduction method for further feature fusion [30]. SR-based methods extract image features and improve the performance of image fusion by learning an overcomplete dictionary [31].

Moreover, owing to the robust capabilities for feature extraction and representation inherent in deep neural networks, more refined and effective image fusion results are realized. Based on the specific architecture and operational principles of the network, deep learning based networks include three main categories, namely, convolutional neural network (CNN) [32], auto-encoder (AE) [33] and generative adversarial network (GAN) [34] based methods. The CNN-based method are used to generate impressive fusion results through specially designed feature extraction networks and reconstruction networks [35]. The AE learns an efficient representation of the source image through an encoder and reconstructs it through a decoder to generate

a complete fused image [36]. In the GAN-based method, the generator is tasked with creating a fused image, and the discriminator ensures that this image encapsulates richer feature information from the source by comparing it directly [37].

Although most existing algorithms produce relatively favorable fusion results, certain limitations remain to be addressed. 1) many traditional methods, such as those based on MST [25] or SR [27], rely on handcrafted fusion rules that may not effectively enhance salient features while preserving texture details. 2) Existing fusion methods obtain satisfactory fused images by designing complex fusion networks, however, the enhancement of salient features is neglected while ensuring the richness of feature information. As shown in Fig 1, both SDNet [32] and ICAFusion [38] result in a fused image that contains complete information, but falls short in highlighting the salient features of thermal radiation, as shown in the purple box. 3) Deep learning-driven fusion methods often require extensive datasets of high-quality natural images to train networks, aiming to make the fusion model have excellent fusion performance and robustness. These limitations highlight the need for a new approach that can effectively enhance salient features without relying on large-scale training data.

To tackle the aforementioned challenges, we presents a new image fusion method based on sub-window variance filter (SVF) and weighted least squares optimization (WLSO). First, in order to more comprehensively separate the salient features, texture and background information into different band layers, a decomposition scheme based on a SVF is proposed to separate the attribute features of different scales into the detail and base layers, respectively. Next, a saliency map measure scheme based on WLSO is designed to compute the weight map of important information in the source image. Then, the obtained weight maps are weighted and summed at the detail and base layers, respectively, to further enhance and highlight the displayability of different features. Finally, feature reconstruction is performed by pixel-level summation to obtain the desired fusion result.

Specifically, the primary objective of this study is to address the limitations of existing fusion methods, particularly the inadequate enhancement of salient features and the reliance on large-scale training datasets. Our proposed method aims to effectively separate and highlight salient features while preserving texture details, without requiring extensive training data, thereby improving the quality and interpretability of infrared and visible image fusion.

The primary contributions of this paper are summarized as follows:

(1) A decomposition method based on SVF is proposed. The significant and structural features in the input image are separated by calculating and comparing the global variance and local variance, so as to effectively decompose different modal features into detail and base layers.

(2) A visual saliency map measure scheme based on WLSO is proposed for different modal features. By calculating the feature intensity distribution of the global region in the source image and then optimizing it using the weighted least squares method, the pixel intensity weight map is obtained.

(3) We conducted extensive comparative experiments with nine advanced fusion methods on three publicly available datasets. The experimental results confirm that the proposed method excels in highlighting significant thermal radiation features while preserving texture details.

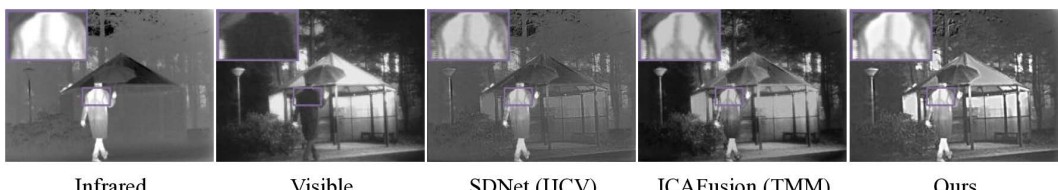

| Infrared | Visible | SDNet (IJCV) | ICAFusion (TMM) | Ours |

**Fig 1. Visualization of fusion results for partial methods.**

In short, the innovation of our study lies in the combination of the SVF for decomposition and the WLSO for saliency map measurement. This combination allows our method to effectively separate and enhance salient features while preserving texture details, addressing the limitations of existing methods that either neglect salient feature enhancement or rely on large-scale training datasets.

The subsequent sections of this paper are structured as follows. Section 2 presents an overview of conventional and deep learning-driven image fusion methodologies, followed by a detailed description of our novel method in Section 3. Comparison experiments and ablation experiments on three datasets are detailed in Section 4 to validate the effectiveness of the proposed method. The conclusion can be found in Section 5. Limitations and future work in Section 6.

## 2. Related work

### 2.1 Conventional image fusion methods

Currently, the mainstream conventional image fusion methods includes MST [3,39], subspace [27,40] and SR [41] -based methods.

In the fusion method based on MST, Tang et al. [3] employed a weighted least squares filter to separate the input image into the base and detail layers, and designed a saliency map measurement based on an adaptive weight assignment strategy and a SVF to enhance the fused base and detail layers, respectively, and ultimately output the fused results with significant gradient features. Liu et al. [42] have extracted infrared target features and texture details from the original images by performing coarse-scale decomposition on infrared images and fine-scale decomposition on visible images, respectively. In addition, Zhang et al. [39] proposed a two-scale decomposition framework based on an average filter to effectively preserve the gradient variations and the thermal radiation features. Li et al. [43] proposed a fusion method (IVFusion) based on MST and paradigm optimization, which takes the pre-fused image as a reference and introduces structural similarity to assess the validity of the detail information, combined with L2 norm optimization method in order to generate the fused image.

In subspace-based fusion methods, typically, the proposed FPDE [26] utilizes principal component analysis (PCA) to fuse high-frequency detail information, which provides ideas for subsequent research. Subsequently, Liu et al. [44] used an improved tensor robust PCA to downscale the image to different subspace regions to achieve effective separation of different modal features in the source image. In addition, Omer et al. [40] captured image salient features via independent component analysis (ICA) and combined it with Chebyshev polynomial approximation to remove the noise, which effectively suppresses the introduction of extraneous details, ensuring that the fused result is free from redundant information from the source image. In their study [45], Mou et al. applied nonnegative matrix factorization (NMF) for feature extraction, effectively enhancing the salient features while retaining the essential texture details.

In the SR-based method, building upon convolutional SR, Wohlberg et al. [41] introduced an image fusion method that offers greater flexibility and efficiency in fusion processes. Aishwarya et al. [27] added supervised dictionary learning to the SR model, which effectively reduced the number of dictionaries trained. To obtain effective information from the source images, Li et al. [46] proposed a fusion method based on latent low-rank representation. However, this method faces significant computational efficiency issues due to the extensive mathematical calculations required to obtain the optimal solution. In addition, Li et al. [47] combined online robust dictionary learning with a guided filter, and used the idea of patch-based clustering to classify similar feature pixels, which achieved to retain the critical information in the input image.

### 2.2. Deep learning fusion methods

Over the past decade, deep neural networks have been increasingly utilized by researchers for their robust feature extraction and representation capabilities [48–51]. Nowadays, research on image fusion algorithms based on deep learning architectures is also gradually becoming a new research hotspot. Depending on the differences in network

architecture, fusion methods based on deep learning are categorized into three main classes, i.e., fusion methods based on CNN [32,35], AE [33], and GAN [52].

In the CNN-based method, Hou et al. [35] designed an unsupervised end-to-end IVIF network with a structure consisting of cascaded dense blocks, and achieved feature reconstruction with stacked convolution operations to generate fusion results. The method effectively and adaptively fuses thermal radiation features and texture details with simultaneous suppression of noise interference. Zhao et al. [53] proposed a algorithm for unfolded image fusion (AUIF), which is a model that contains two encoders and a decoder, and efficiently generates a fused image that not only captures salient targets but also preserves clear and discernible detail. Furthermore, aiming to accomplish an image fusion task that is not only more efficient but also highly accurate, Liu et al. [54] proposed a convolution-based lightweight pixel-level unified fusion network. PMGI utilizes multi-gradient information to guide the image fusion process and employs CNN to extract and fuse multi-scale features [55].

In the AE-based method, Li et al. [36] designed dense blocks as encoder structures for the first time, which effectively improved the information flow in the network and prevented information loss. Additionally, Li et al. [56] have pioneered the incorporation of an intricate nested connection architecture within fusion networks, called NestFuse. The nested architecture adequately preserves and utilizes features at different scales, and a fusion strategy based on spatial attention and channel attention models is designed for fusing multi-scale depth features. Building upon NestFuse, Li et al. [57] proposed a learnable residual fusion network to replace manually designed fusion strategies, effectively addressing the issue of insufficient information utilization. Xing et al. [58] introduced a compressed fusion network named CFNet, which incorporates the idea of image compression based on variational autoencoders to achieve joint optimization of image fusion and data compression.

In the GAN-based approach, Ma et al. [37] proposed a fusion network with dual discriminators, which effectively solved the problem of the lack of infrared image information. Moreover, Li et al. [52] proposed a fusion network that combines the attention mechanism with a GAN, called AttentionFGAN. In this method, the information within the input image is retained to the maximum extent, with key details being effectively brought into focus. Le et al. [59] first proposed a fusion network of continuously learned GAN, named UIFGAN, for a unified image fusion task. To tackle the problem of global information loss, Wu et al. [60] proposed a globally aware generative adversarial fusion network and introduced differential image loss to further constrain the generator to learn important information from the source images.

## 3. Methodology.

We initially offer a comprehensive overview of the proposed methodology. Then, we describe in detail the proposed decomposition scheme and the concrete realization process of the fusion strategy.

### 3.1. Overall overview

As illustrated in Fig 2, the overall framework of the proposed method is systematically displayed. The specific flow of the proposed method is as follows:

(1) Image Decomposition

Due to the richness and diversity of the information contained in the source images. Therefore, we propose an image decomposition scheme based on SVF [61] aiming at separating different feature information into different band layers. The base layer contains the overall contrast and background features, while the detail layer contains texture and edge information. The decomposition process is given by:

$$\left( \left\{ B_{ir}, B_{vis} \right\}, \left\{ D_{ir}, D_{vis} \right\} \right) = SVF \left( \left\{ I_{ir}, I_{vis} \right\} \right)$$

(1)

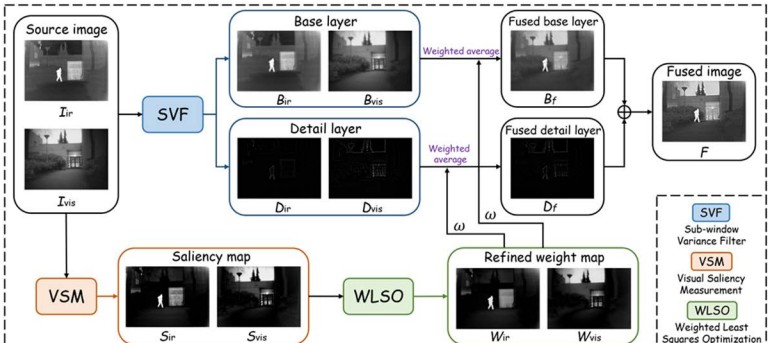

**Fig 2. The overall pipeline of the method proposed in this paper.**

where $B_{ir}$ and $B_{vis}$ are the base layer, $D_{ir}$ and $D_{vis}$ are the detail layer, SVF($\cdot$) denotes the SVF operator, and $I_{ir}$ and $I_{vis}$ denote the infrared image and the visible image, respectively. To ensure clarity, the SVF operator is explained in detail in *Section 3.2*, including the mathematical formulation and implementation steps.

(2) Visual Saliency Map and Weight Calculation

In order to make the key information in the source image can be fully preserved and fully presented in the fusion result. We designed a visual saliency map measure scheme based on WLSO. The whole process is represented by the following steps:

*Step 1*: Compute the visual saliency maps for the infrared and visible images using the Visual Saliency Maping (VSM) operator:

$$\{S_{ir}, S_{vis}\} = VSM\left(\{I_{ir}, I_{vis}\}\right) \tag{2}$$

*Step 2*: Optimize the saliency maps using Weighted Least Squares Optimization (WLSO) to obtain the weight maps:

$$\{W_{ir}, W_{vis}\} = WLSO\left(\{S_{ir}, S_{vis}\}\right) \tag{3}$$

*Step 3*: Calculate the final weight coefficients:

$$\omega = 0.5 + \frac{W_{ir} - W_{vis}}{2} \tag{4}$$

where $S_{ir}$ and $S_{vis}$ denote the saliency maps of the source images, respectively. *VSM*($\cdot$) denotes the operator for computing the visual saliency maps. $W_{ir}$ and $W_{vis}$ denote the weight maps after weighted least squares optimization, respectively. This ensures that the fusion process adaptively balances the contributions of the infrared and visible images based on their saliency information. *WLSO*($\cdot$) denotes the weighted least squares optimization operator. $\omega$ denotes the final weight coefficients. The computation of the saliency maps and the optimization process are further elaborated in *Section 3.3*.

(3) Fusion of Base and Detail Layers

The obtained weight maps are weighted and summed with the base and detail layers, respectively, with the purpose of fully integrating the private modal information in the base and detail layers. This process is denoted as:

$$B_f = \omega \cdot B_{ir} + (1 - \omega) \cdot B_{vis} \tag{5}$$

$$D_f = \omega \cdot D_{ir} + (1 - \omega) \cdot D_{vis} \tag{6}$$

where $B_f$ and $D_f$ denote the fused base and detail layer, respectively.

(4) Final Fusion Result

The fused base and detail layers are summed at the element level to obtain the desired result $F$, denoted as:

$$F = B_f + D_f \tag{7}$$

## 3.2. Decomposition method

The sub-window variance filter (SVF) is an edge-preserving filter constructed using local edge statistics information, which improves the edge-awareness of the filter by computing the information of the spatial statistics of the image [61]. The SVF-based decomposition scheme separates the source image into base and detail layers. Specifically, the result of the SVF can be viewed as a linear combination of the input image and its smoothed filtered result. Here, it is assumed that the image patch of the input image centered at pixel $k$ is $I_k$, and the result after the sub-window variance filter is $I'_k$, which can be expressed as:

$$I'_k = C_k I_k + (1 - C_k) F(I_k) \tag{8}$$

where the weight coefficient $C_k$ is used to control the contribution of the input image block $I_k$ in the filtered image $I'_k$, and $C_k \in [0, 1]$. $F(\cdot)$ is a generalized box filter.

After Eq (7), it is able to limit the pixel value of the filtered image $I'_k$ between the pixel value and the mean value of the original image, thus effectively eliminating the occurrence of oversharpening in the bilateral filter [28]. Since F($\cdot$) is a box filter, Eq. (7) can be further rewritten as:

$$I'_k = C_k I_k + (1 - C_k) \left( \frac{1}{|\omega|} \sum_{i \in \omega_k} p_i \right) \tag{9}$$

where $\omega_k$ denotes the receptive domain of the filter when centered at pixel $k$, and $p_i$ denotes the intensity value of the $i$th pixel in the receptive domain.

The computation of weighting factor $C_k$ is explicitly described as follows: Divide the filtering region $W$ into four equal-sized sub-windows $A$, $B$, $C$, and $D$. This division is based on the need to capture local variance information effectively in different spatial directions (e.g., horizontal, vertical, and diagonal). The choice of four sub-windows provides a balanced representation of local edge statistics, ensuring that the filter can adaptively preserve edge features while smoothing homogeneous regions. Calculate the set of local variance values corresponding to the four subregions $V = \{\sigma_A^2, \sigma_B^2, \sigma_C^2, \sigma_D^2\}$, respectively. As well as the global variance value $\sigma_W^2$ of the region to be filtered $W$. Then the weighting factor $C_k$ can be defined as:

$$C_k = \min\left(1, \frac{\sigma_{max}^2}{\sigma_{min}^2 + \phi}\right) \tag{10}$$

where $\sigma_{max}^2 = max\left(\{\sigma_W^2, V\}\right)$ and $\sigma_{min}^2 = min(V)$. $\phi$ is a regularization parameter. Here, when $C_k < 1$, then $I'_k = C_k I_k + (1 - C_k) F(I_k)$, and edge features will be mostly preserved. When $C_k = 1$, then $I'_k = I_k$ and the edge is fully preserved. Moreover, when $C_k$ is a minimal value such that $I'_k \approx F(I_k)$, most of the region in the image is smoothed.

Based on the above description, from Eqs. (8) to (10), the operator $SVF(\cdot)$ for the sub-window variance filter can be expressed as:

$$B = SVF(I) \tag{11}$$

where $B$ denotes the base layer which contains most of the background features and overall contrast information. Texture details and edge features, which constitute the high-frequency information in an image, can be precisely extracted by calculating the differences between the original and the corresponding base layer. In other words, The detail layer D is extracted by subtracting the base layer from the original image, denoted as:

$$D = I - B \tag{12}$$

where $D$ contains high-frequency information such as texture and edges.

### 3.3 Fusion strategy

First, the visual saliency map measure scheme is a classical saliency map computation that aims to establish the contrast intensity present between different pixels in an image to define the saliency map [62]. Assuming that the intensity value of the $i$th pixel in the input image $I$ is denoted as $I_i$, the saliency map $S(i)$ of pixel $i$ is denoted as:

$$S(i) = \left| I_i - I_1 \right| + \left| I_i - I_2 \right| + \ldots + \left| I_i - I_N \right| \tag{13}$$

where $N$ stands for the total number of pixels and $|\cdot|$ indicates the absolute operator. Eq. (13) can be further rewritten as:

$$S(i) = \sum_{j=1}^{N} \left| I_i - I_j \right| \tag{14}$$

Then, $S(i)$ is normalized to $[0, 1]$ ensures that the saliency values are comparable across different images. Eq. (14) that represents the $VSM(\cdot)$ operator.

To further clarify the fusion strategy, the optimization process using the weighted least squares method is described in detail.

Specifically, in order to further enable the features on the saliency map to be mapped to the fusion results and to exclude potential redundant information and noise interference, for this purpose, we have optimized the saliency map using a weighted least squares method [28]. We take the salient map $S$ as input, and the optimization function can be expressed as,

$$f(S) = \sum_{p} \left( (W_p - S_p)^2 + \lambda \left( \psi_{x,\,p}(S) \left( \frac{\partial w}{\partial x} \right)_p^2 + \psi_{y,\,p}(S) \left( \frac{\partial w}{\partial y} \right)_p^2 \right) \right) \tag{15}$$

The optimization function in Eq. (15) aims to minimize the difference between the saliency map $S$ and the weight map $W$, while penalizing large gradients in $W$. This ensures that the weight map is smooth and free from noise. In Eq. (15), where $\lambda$ denotes the balance coefficient, which controls the trade-off between fidelity to the saliency map and smoothness of the weight map. The choice of $\lambda$ is based on ablation experiments to achieve optimal performance, $p$ denotes the spatial location of the pixel, and $\psi_x$ and $\psi_y$ denote the weight matrices of the gradient in the $x$ and $y$ directions, respectively. Next, Eq. (15) can be converted to matrix form as follows.

$$f(S) = (W - S)^T (W - S) + \lambda \left( W^T H_x^T A_x H_x W + W^T H_y^T A_y H_y W \right) \tag{16}$$

where $A_x$ and $A_y$ are diagonal weight matrices and $H_x$ and $H_y$ are matrix representations of discrete difference operators. Let the derivative of $f(S)$ be zero, i.e.,

$$f'(S) = 2W - 2S + 2\lambda \left( H_x^T A_x H_x + H_y^T A_y H_y \right) W = 0 \tag{17}$$

According to Eq. (17), the solution representation of the optimization goal can be obtained as:

$$(E + \lambda L)W = S\# \tag{18}$$

where $E$ is the unit matrix and $L = H_x^T A_x H_x + H_y^T A_y H_y$. From Eq. (18), the operator $WLSO(\cdot)$ based on weighted least squares optimization can be expressed as:

$$W = WLS(S) = (E + \lambda L)^{-1} S \tag{19}$$

The optimized weight maps $W_{ir}$ and $W_{vis}$ are used to compute the final weight coefficients $\omega$ (Eq. 4), which are then applied to fuse the base and detail layers (Eqs. 5 and 6) to obtain the final fused image $F$ (Eq. 7).

## 4. Experiments

### 4.1. Experimental settings

We conducted extensive comparative experiments, both qualitatively and quantitatively, on the TNO [63], RoadScene [64], and LLVIP [65] datasets to demonstrate the effectiveness and robust fusion capabilities of our proposed method. Then, nine state-of-the-art (SOTA) image fusion methods are selected for comparison, including SDNet [32], ICAFusion [38], GANMcC [66], SwinFusion [67], PIAFusion [2], TarDAL [68], TGFuse [69], CrossFuse [70] and SeAFusion [71]. In the quantitative experiments, six pivotal evaluation metrics were chosen to assess the fusion performance comprehensively, including spatial frequency (SF) [72], mean squared error (MSE) [73], correlation coefficient (CC) [74], sum of the correlations of differences (SCD) [75], structural similarity index measure (SSIM) [76], peak signal to noise ratio (PSNR) [77], and Qabf [78]. All the experiments are implemented on a computer with NVIDIA GeForce RTX 3060 GPU and 16 GB memory.

### 4.2. Comparison experiments on the TNO dataset

First, Figs 3 and 4 give some of the visualization results on the TNO dataset. As shown in Fig 1, SwinFusion, PIAFusion, and SeAFusion cannot effectively retain the background brightness, resulting in the sky about the shape of the cloud features can not be completely presented. This is likely due to their insufficient ability to balance the intensity information between infrared and visible images during fusion, leading to the loss of low-frequency background details. TarDAL introduces some of the background noise. This noise may stem from its reliance on dense feature extraction, which can amplify irrelevant details in the background. In addition, SDNet, GANMcC, and SwinFusion are deficient in preserving significant edge features in visible images, as illustrated by the green boxes. In contrast, our method effectively avoids the above mentioned problems. This is attributed to our WLSO-optimized saliency maps, which ensure that both low-frequency background and high-frequency edge features are preserved during fusion. In Fig 2, all SOTA methods result in a complete fused image. However, in terms of details, the proposed method exhibits superior performance in enhancing texture details, allowing for a clearer depiction of fine elements such as tree branches, as indicated in the blue boxes.

Furthermore, the average values of six complementary metrics are listed in Table 1. Significantly, our approach demonstrates superior performance across all metrics, and one more metric achieves the second best performance. The best SF and Qabf illustrates that our fusion results achieves the most refined sharpness, which is a direct result of our emphasis on preserving high-frequency details. The best MSE also illustrates that our fusion results have better image quality. This is because our method minimizes information loss during fusion by effectively combining the base and detail layers. The excellent performance of SCD shows that our method fully complements the information in the source image. This is achieved by our visual saliency map strategy, which ensures that complementary information from both modalities is

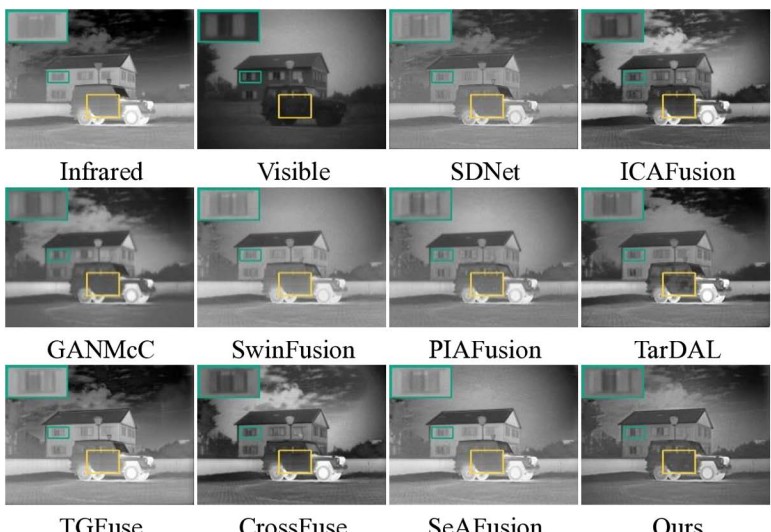

**Fig 3. Visualization results of 10 methods on "*Marne_04*" image pairs in the TNO dataset.**

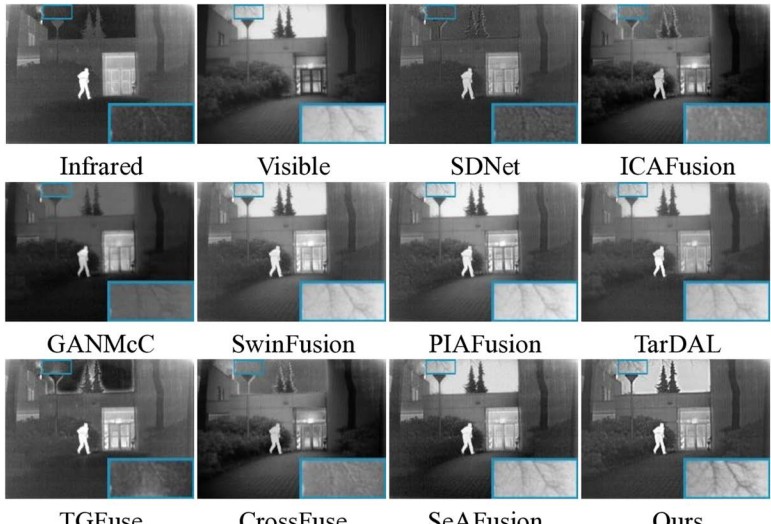

**Fig 4. Visualization results of 10 methods on "*Kaptein_1123*" image pairs in the TNO dataset.**

**Table 1. Results of quantitative comparisons on the TNO dataset, where bold font indicates best and underlining indicates second best.**

|  | SDNet | ICAFusion | GANMcC | SwinFusion | PIAFusion | TarDAL | TGFuse | CrossFuse | SeAFusion | Ours |
|---|---|---|---|---|---|---|---|---|---|---|
| SF↑ | 0.0457 | 0.0471 | 0.0242 | 0.0420 | 0.0440 | 0.0425 | 0.0411 | 0.0479 | 0.0482 | **0.0503** |
| MSE↓ | 0.0493 | 0.0491 | 0.0547 | 0.0620 | 0.0580 | 0.0466 | 0.0545 | 0.0485 | 0.0595 | **0.0373** |
| CC↑ | 0.4562 | 0.4362 | **0.5219** | 0.4704 | 0.4576 | 0.4493 | 0.4211 | 0.4284 | 0.4781 | 0.4986 |
| SCD↑ | 1.5654 | 1.5112 | 1.7075 | 1.7098 | 1.6993 | 1.5844 | 1.5304 | 1.4716 | 1.7260 | **1.7671** |
| SSIM↑ | 0.9344 | 0.8626 | 0.8788 | 0.9345 | 0.9208 | 0.8846 | 0.9156 | 0.8637 | 0.9139 | **0.9452** |
| PSNR↑ | 62.1044 | 61.7979 | 61.5138 | 61.0565 | 61.3149 | 62.4008 | 61.9568 | 61.8780 | 61.3631 | **63.3296** |
| Qabf↑ | 0.2987 | 0.3014 | 0.3329 | 0.3991 | 0.4125 | 0.4239 | 0.3872 | 0.4019 | 0.4689 | **0.4730** |

integrated seamlessly. In addition, the proposed method achieves the best SSIM and PSNR, indicating superior structural similarity and contains less noise information.

### 4.3. Comparison experiments on the RoadScene dataset

Fig 5 gives some visualization of the fusion results in the RoadScene dataset. It is worth noting that SDNet, ICAFusion, GANMcC, TGFuse, and CrossFuse are unable to display the background information correctly, and all methods are prone to different levels of spectral interference in the infrared imager, which is particularly evident in the discoloration of the sky background. This issue arises because these methods do not adequately handle the intensity differences between infrared and visible images, leading to improper fusion of low-frequency background regions. Furthermore, SwinFusion, PIAFusion, and TarDAL do not effectively retain critical detailed features in infrared images, as shown in the green box. On the other hand, SDNet, GANMcC, and TGFuse, in particular, are affected by the noise of the infrared image, which results in critical features not being effectively fused, and there is insufficient information retention. In contrast, our method effectively solves the above problem. This is achieved by our SVF-based decomposition, which separates noise from meaningful information. Our fused result not only correctly displays the background information, but also shows the outline of the cable completely.

Similarly, Table 2 gives the results of the quantitative comparison of the 10 methods on the RoadScene dataset. It can be observed that our methods still have a competitive performance on most of the metrics. Attributed to visual saliency maps optimized by WLSO, which enables the base and detail layers to prevent information loss when key features are fused, and the maximum MSE further reflects this advantage. Also, the maximum PSNR similarly reflects that our fusion results are characterized by sharper texture details. This is because our method effectively preserves high-frequency

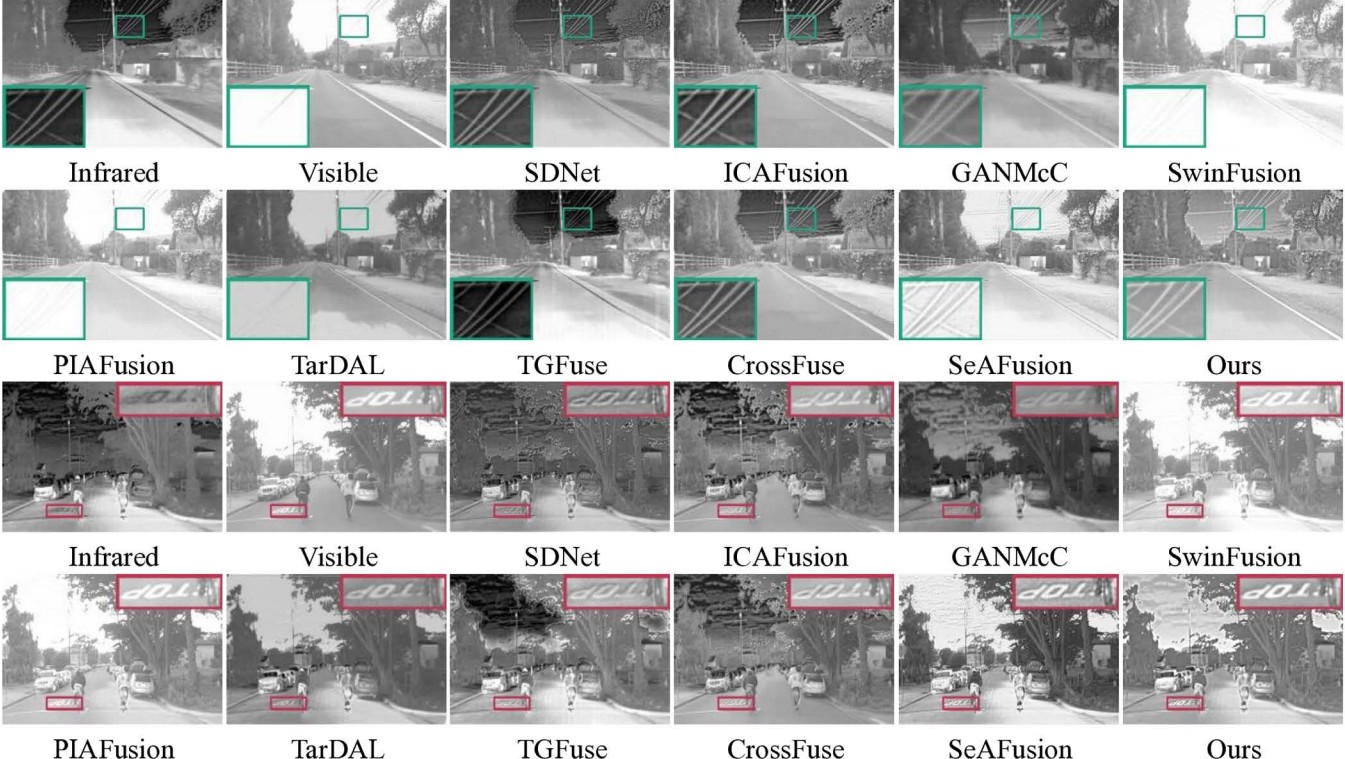

**Fig 5. Visualization results of 10 methods in the RoadScene dataset.**

**Table 2. Quantitative comparison results on the RoadScene dataset.**

|  | SDNet | ICAFusion | GANMcC | SwinFusion | PIAFusion | TarDAL | TGFuse | CrossFuse | SeAFusion | Ours |
|---|---|---|---|---|---|---|---|---|---|---|
| SF↑ | 0.0630 | 0.0583 | 0.0364 | 0.0489 | 0.0508 | 0.0467 | 0.0602 | 0.0600 | **0.0775** | 0.0634 |
| MSE↓ | 0.0672 | 0.0699 | 0.0762 | 0.0865 | 0.0837 | 0.0553 | 0.0692 | 0.0686 | 0.0687 | **0.0452** |
| CC↑ | 0.4345 | 0.4100 | **0.5245** | 0.4960 | 0.4498 | 0.4536 | 0.3694 | 0.3966 | 0.4905 | 0.5112 |
| SCD↑ | 1.6719 | 1.4403 | **1.7996** | 1.6535 | 1.5654 | 1.4518 | 1.2741 | 1.3938 | 1.6548 | 1.7098 |
| SSIM↑ | **0.9382** | 0.7663 | 0.8655 | 0.8414 | 0.8046 | 0.8185 | 0.9284 | 0.7530 | 0.8765 | 0.9086 |
| PSNR↑ | 60.1372 | 59.9011 | 59.5318 | 59.0478 | 59.2409 | 60.9493 | 60.1478 | 60.0161 | 60.0771 | **61.9162** |
| Qabf↑ | 0.4398 | 0.4173 | 0.3972 | 0.5524 | 0.4419 | 0.3987 | 0.4331 | 0.4120 | **0.5877** | 0.5791 |

details through the detail layer fusion process. The second-ranked Qabf illustrates the advantages of our method in preserving scene edge features and reflecting information richness. In conclusion, the comprehensive comparative results reveals that our method is able to achieve excellent fusion results.

## 4.4. Comparison experiments on the LLVIP dataset

As shown in Fig 6, we actively engaged in comparative experiments on the LLVIP dataset to showcase the superior fusion performance of our proposed method with infrared-RGB visible images. In daytime scene, the significant features of thermal radiation in the infrared image can provide more complete and complementary information for the visible image. However, SDNet suffers from the noise of infrared images, which makes the fusion quality inferior. In addition, GANMcC

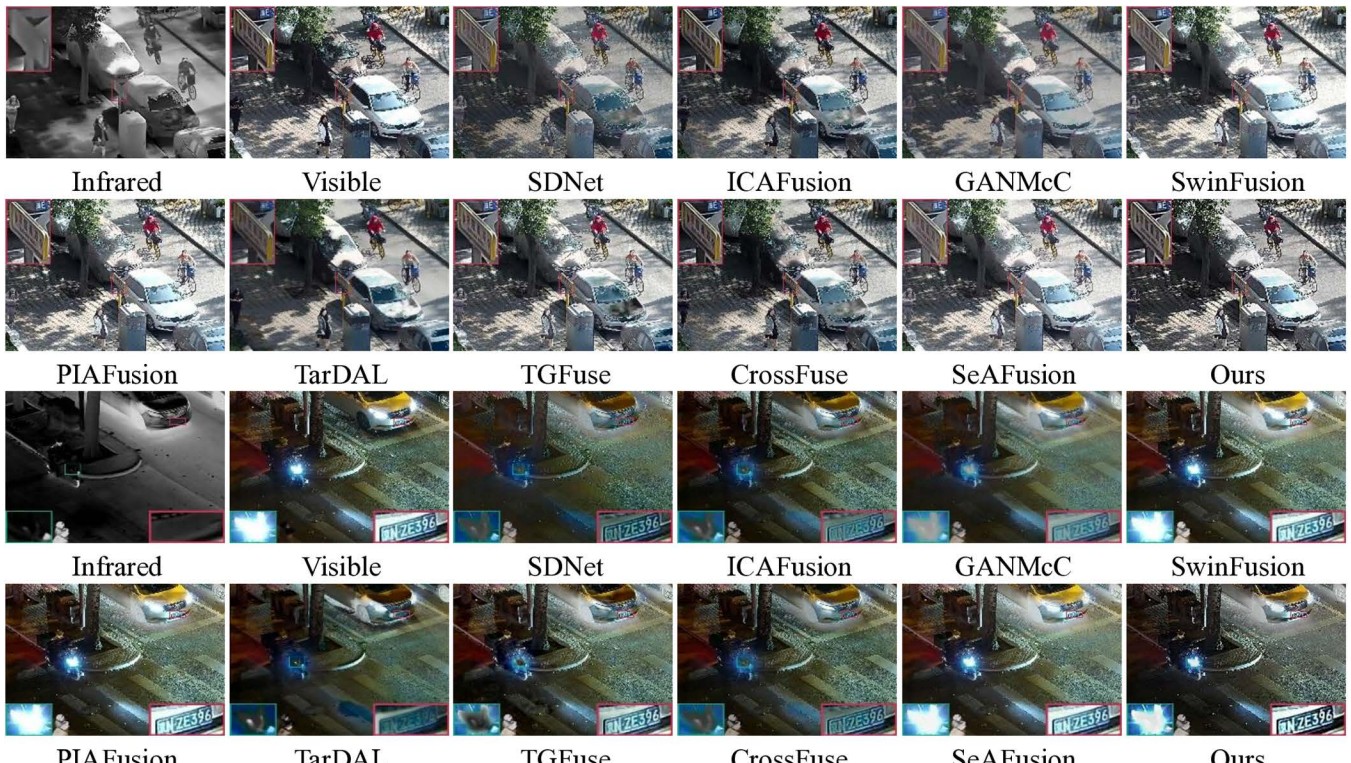

**Fig 6. Qualitative comparison of the 10 methods on images from the LLVIP dataset.**

and TarDAL are deficient in preserving detailed features. This deficiency arises because these methods do not adequately emphasize high-frequency detail preservation during fusion. In nighttime scenes, the fused images are often susceptible to varying degrees of spectral contamination from the visible images, especially SDNet, ICAFusion, TarDAL, TGFuse, and CrossFuse, as shown in the green box. On the other hand, ICAFusion and TarDAL are also unable to completely preserve the license plate information. This is likely due to their insufficient ability to retain fine details during fusion. In contrast, our method and SeAFusion are able to completely overcome the above problems by effectively synthesizing significant thermal radiation features and texture information, without being affected by lighting conditions.

Furthermore, Table 3 offers a comprehensive display of the objective metrics values. Obviously, our method obtains the second-best ranking performance on the majority of the metrics. Attributed to the effective SVF-based decomposition and the visual saliency measure strategy based on WLSO, the proposed method effectively preserves the texture details while highlighting the salient information, as reflected in metrics SF, CC and Qabf. From the comparative experiments, it is evident that our proposed method is characterized by its favorable fusion performance and robustness.

## 4.5. Ablation experiment

### 4.5.1. Effectiveness analysis of decomposition method and fusion strategy.
Ablation experiments were implemented to systematically evaluate the impact of the decomposition method and fusion strategy on the overall effectiveness of our approach. First, for the decomposition method, we chose the classical FPDE as an alternative to SVF, as shown in the group of Fig 7 (a). However, the fusion results obtained using the FPDE-based decomposition scheme are deficient in preserving texture details. In contrast, attributed to SVF for determining salient features by calculating pixel variance allow salient gradient features to be preserved. Next, for the fusion strategy, we chose a simple weighted average approach as an alternative to the WLSO-based saliency measure scheme, and the fusion results are shown in group (b) of Fig 7. It is evident that our method possesses a notable advantage in emphasizing the thermal radiation

**Table 3. Quantitative comparison results on the LLVIP dataset.**

|  | SDNet | ICAFusion | GANMcC | SwinFusion | PIAFusion | TarDAL | TGFuse | CrossFuse | SeAFusion | Ours |
|---|---|---|---|---|---|---|---|---|---|---|
| SF↑ | 0.0483 | 0.0490 | 0.0313 | **0.1025** | 0.0618 | 0.0259 | 0.0566 | 0.0551 | 0.0556 | 0.0691 |
| MSE↓ | 0.0281 | 0.0279 | **0.0245** | 0.0366 | 0.0339 | 0.0384 | 0.0381 | 0.0276 | 0.0400 | 0.0251 |
| CC↑ | 0.6547 | 0.6800 | **0.7176** | 0.6761 | 0.6780 | 0.6427 | 0.6500 | 0.6617 | 0.6795 | 0.6961 |
| SCD↑ | 1.0168 | 1.2231 | 1.3799 | **1.6451** | 1.5351 | 0.9705 | 1.4463 | 1.2296 | 1.6027 | 1.4569 |
| SSIM↑ | 0.9533 | 0.9245 | 0.9119 | 0.9258 | **0.9727** | 0.8372 | 0.9601 | 0.9210 | 0.9634 | 0.9698 |
| PSNR↑ | 63.8071 | 63.7798 | **64.3269** | 62.6638 | 62.9939 | 62.5021 | 62.5094 | 63.8184 | 62.2536 | 64.2781 |
| Qabf↑ | 0.5329 | 0.5271 | 0.3895 | 0.5572 | **0.6120** | 0.2334 | 0.4968 | 0.5113 | 0.4732 | 0.6019 |

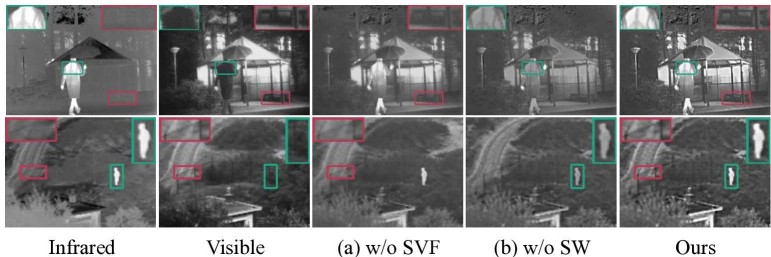

Infrared          Visible          (a) w/o SVF          (b) w/o SW          Ours

**Fig 7. Qualitative comparison results of ablation experiments on the TNO dataset, where (a) w/o SVF denotes the use of FPDE as the decomposition method and (b) w/o SW denotes the use of average method as the fusion strategy.**

   

features, attributed to the superiority of the visual saliency measure in extracting the salient regions, and then after the WLSO, the resulting saliency weight map can more completely reflect the salient features in the region of interest. Moreover, Table 4 presents the objective comparison values obtained by performing the above ablation experiments on the TNO dataset. The results clearly indicate that our method holds an advantageous position in the majority of the evaluated metrics.

**4.5.2. Parameter sensitivity analysis.** In addition, we conducted relevant ablation experiments on the choice of the balance coefficient $\lambda$ in WLSO, aiming to optimize the performance of the proposed method. We performed the relevant experiments on the TNO dataset and the quantitative results are shown in Table 5. It can be found that, as $\lambda$ increases, SF slightly decreases, indicating a reduction in high-frequency details. This is expected because a larger $\lambda$ emphasizes smoothness in the weight map, which may suppress some fine details. MSE decreases as $\lambda$ increases, indicating better fidelity to the source images. This is because a larger $\lambda$ reduces noise and artifacts in the weight map, leading to more accurate fusion results. CC increases with $\lambda$, this is consistent with the improved fidelity observed in the MSE results. SCD decreases slightly as $\lambda$ increases, indicating a more balanced integration of information from the source images. SSIM remains relatively stable across different values of $\lambda$, with a slight improvement at higher values. PSNR increases with $\lambda$, indicating better noise suppression and higher fidelity to the source images. Qabf increases slightly as $\lambda$ increases, indicating better preservation of information from the source images. Based on the experimental results, we use $\lambda =1$ as the default value in this paper, as it provides a good balance between smoothness and fidelity, achieving high performance across multiple evaluation metrics.

## 4.6 Analysis of running efficiency

Running efficiency is one of the important indicators for evaluating the performance of a method. Therefore, we conducted comparative experiments on running efficiency between our proposed method and nine advanced methods on the TNO dataset, and the results are listed in Table 6. It is evident that, due to the lightweight network architecture design and GPU acceleration, the deep learning-based method SeAFusion achieved a significant advantage. Additionally, CrossFuse ranked second. In contrast, our method, based on traditional optimization theory, requires extensive mathematical calculations to compute the optimal solution, and thus does not exhibit particularly outstanding performance in computational efficiency. Therefore, in future research, how to effectively reduce the iterative computation time will be an area that needs further exploration.

**Table 4. Results of objective metrics on ablation studies on the TNO dataset.**

|          | SF↑    | MSE↓   | CC↑    | SCD↑   | SSIM↑  | PSNR↑   | Qabf↑  |
|----------|--------|--------|--------|--------|--------|---------|--------|
| w/o SVF  | 0.0313 | 0.0357 | 0.4834 | 1.7311 | 0.9232 | 63.2522 | 0.4227 |
| w/o SW   | 0.0303 | **0.0323** | **0.5253** | 1.6443 | 0.9361 | **64.0788** | 0.4121 |
| Ours     | **0.0503** | 0.0373 | 0.4986 | **1.7671** | **0.9452** | 63.3296 | **0.4807** |

**Table 5. Fusion performance under different balance coefficient values $\lambda$.**

| $\lambda$ | SF↑    | MSE↓   | CC↑    | SCD↑   | SSIM↑  | PSNR↑   | Qabf↑  |
|------|--------|--------|--------|--------|--------|---------|--------|
| 0.01 | **0.0429** | 0.0338 | 0.5526 | **1.7484** | 0.9256 | 62.8410 | 0.3971 |
| 0.1  | 0.0418 | 0.0336 | 0.5527 | 1.7478 | **0.9257** | 62.8627 | 0.3958 |
| 1    | 0.0405 | 0.0329 | 0.5532 | 1.7446 | 0.9236 | 62.9528 | 0.3971 |
| 3    | 0.0399 | 0.0322 | 0.5542 | 1.7412 | 0.9238 | 63.0507 | 0.3989 |
| 5    | 0.0396 | 0.0317 | 0.5550 | 1.7392 | 0.9242 | 63.1153 | 0.3999 |
| 7    | 0.0394 | **0.0314** | **0.5558** | 1.7377 | 0.9244 | **63.1662** | **0.4000** |

**Table 6. Results of runtime comparison for 40 pairs of images on the TNO dataset. (Unit: seconds).**

| Method | SDNet | ICAFusion | GANMcC | SwinFusion | PIAFusion |
|---|---|---|---|---|---|
| Times | 0.2013 | 0.2457 | 2.4002 | 1.9245 | 0.9351 |
| Method | TarDAL | TGFuse | CrossFuse | SeAFusion | Ours |
| Times | 1.2104 | 0.1722 | 0.1328 | **0.0054** | 1.9433 |

## 5. Conclusion

In this paper, we proposed a novel IVIF method based on SVF and WLSO, effectively separating salient features and texture details while enhancing their visibility in fused images. Experimental results demonstrate superior performance over nine advanced methods, particularly in target highlighting and detail preservation. The method's independence from large-scale training datasets is a significant advantage, reducing computational costs and avoiding distortion issues.

## 6. Limitations and future work

Despite the promising results achieved by our proposed method, several limitations should be noted. First, while our method performs well on standard datasets, it may face challenges in highly complex scenes where the distinction between salient targets and background textures is less clear. Future work could explore adaptive parameter optimization to improve performance in such scenarios. Second, the applicability of our method to other fusion tasks, such as medical image fusion or multi-spectral fusion, has not been fully explored. Future studies will investigate the generalization capability of our approach across different domains.

## Author contributions

**Writing – original draft:** Peicheng Wang, Tingsong Li, Pengfei Li.

**Writing – review & editing:** Pengfei Li.

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
