## [Decision Letter · Decision Letter 0]

Dear Dr. Li,

Thank you for submitting your manuscript to PLOS ONE. After careful consideration, we feel that it has merit but does not fully meet PLOS ONE’s publication criteria as it currently stands. Therefore, we invite you to submit a revised version of the manuscript that addresses the points raised during the review process.

We look forward to receiving your revised manuscript.

Kind regards,

Xiongkuo Min

Academic Editor

PLOS ONE

Journal Requirements:

4. In the online submission form, you indicated that the data used in this study are available upon request from the corresponding author.

5. We note that Figure 1 and 2 in your submission contain copyrighted images. All PLOS content is published under the Creative Commons Attribution License (CC BY 4.0), which means that the manuscript, images, and Supporting Information files will be freely available online, and any third party is permitted to access, download, copy, distribute, and use these materials in any way, even commercially, with proper attribution. For more information, see our copyright guidelines: http://journals.plos.org/plosone/s/licenses-and-copyright.

a. You may seek permission from the original copyright holder of Figure 1 and 2 to publish the content specifically under the CC BY 4.0 license. 

6. We note that Figure 4 includes an image of a [patient / participant / in the study]. 

7. Please upload a new copy of Figure 1, 2, 3, and 4 as the detail is not clear. Please follow the link for more information: https://blogs.plos.org/plos/2019/06/looking-good-tips-for-creating-your-plos-figures-graphics/"" https://blogs.plos.org/plos/2019/06/looking-good-tips-for-creating-your-plos-figures-graphics/

Reviewers' comments:

Reviewer's Responses to Questions

**Comments to the Author**

1. Is the manuscript technically sound, and do the data support the conclusions?

Reviewer #1: Yes

Reviewer #2: No

Reviewer #3: Yes

Reviewer #4: Yes

2. Has the statistical analysis been performed appropriately and rigorously?

Reviewer #1: Yes

Reviewer #2: No

Reviewer #3: No

Reviewer #4: Yes

3. Have the authors made all data underlying the findings in their manuscript fully available?

Reviewer #1: Yes

Reviewer #2: Yes

Reviewer #3: No

Reviewer #4: Yes

4. Is the manuscript presented in an intelligible fashion and written in standard English?

Reviewer #1: Yes

Reviewer #2: Yes

Reviewer #3: No

Reviewer #4: Yes

Reviewer #1: The article was reviewed under the title "Infrared and visible image fusion method based on sub-window variance filter and weighted least squares optimization".

Overall, the focus of the study is good. However, I would like to offer several recommendations that authors may find useful in the process of revising their manuscript:

1- The abstract is very general, and it is necessary to provide more details in the abstract, given the variety of topics discussed.

2- Considering the classification of Related works into two categories: conventional methods and deep learning, it is better to examine more of the activities carried out in order to compare and show the strengths.

3- Considering the superiority of the model, if the challenges of previous research are examined, it can enrich the effectiveness of the article.

4- The article states that the weight factor (CK) is divided into 4 sub-windows. Are the 4 sub-windows specified by the authors or is it a fixed value?

5- If the authors chose 4 sub-windows, what was the reason for choosing this value?

6- Although the evaluation criteria used are well-known, it is better to include their full text in a footnote the first time their summary is used in the text of the article.

7- In the tables where the evaluation criteria obtained are compared with other studies, in some cases the criteria obtained by other researchers have a slight advantage, but the authors have stated that their model has superiority.

Reviewer #2: 1- The title should be improved.

2- The objectives and the rationale of the study are recommended to be clearly stated.

3- The concluding remarks of the abstract are not well-written. It's merely the repetition of the objectives and title of the manuscript. Please add method limitations and justification to the abstract.

4- The innovation of using this study is not very clear. I do not see a clear reason that this study can perform better than others. Why did the authors choose the method for this study?

5- The necessity & novelty of the manuscript should be presented and stressed in the "Introduction" section.

6- The application/theory/method/study reported is not in sufficient detail to allow for its replicability and/or reproducibility. Therefore, it is suggested to make it clear to show all steps to build the model.

7- The problem statement and gap study are not clear.

8- The method is not clear. Therefore, it must be shown and clarified to show all steps.

9- The interpretation of results and study conclusions are not supported by providing the reasons behind why they show that. Therefore, it is recommended to deepen the discussion.

10- It is recommended to emphasize the strengths of the study clearly.

11- The limitations of the study should be stated.

12- The manuscript structure, flow, or writing needs some improvements.

13- The manuscript is benefit from language editing. The English of the paper is readable; however, I would suggest the authors to have it checked preferably by a native English-speaking person to avoid any mistakes.

14- I noticed that the conclusion section tends to repeat the abstract and results. The conclusion paragraph should be short, impactful, and direct the reader to this research's next steps and opportunities.

15- It will be nice to add some new references to show that your study is updated, such as: Zhou, Zhanxin, and Ruibo Wu. "Stock Price Prediction Model Based on Convolutional Neural Networks." Journal of Industrial Engineering and Applied Science 2.4 (2024): 1-7; Alakbari, Fahd Saeed, et al. "Prediction of critical total drawdown in sand production from gas wells: Machine learning approach." The Canadian Journal of Chemical Engineering 101.5 (2023): 2493-2509.; Alakbari, Fahd Saeed, et al. "Deep learning approach for robust prediction of reservoir bubble point pressure." ACS omega 6.33 (2021): 21499-21513.; Alakbari, Fahd Saeed, et al. "A gated recurrent unit model to predict Poisson's ratio using deep learning." Journal of Rock Mechanics and Geotechnical Engineering 16.1 (2024): 123-135; Zhou, Zhanxin, and Ruibo Wu. "Stock Price Prediction Model Based on Convolutional Neural Networks." Journal of Industrial Engineering and Applied Science 2.4 (2024): 1-7; Wu, Ruibo, Tao Zhang,

Reviewer #3: 1. In the process of introducing the sub - window variance filter (SVF) and weighted least squares optimization (WLSO), the following aspects need to be improved:

- Provide a more detailed explanation of the derivation steps of key formulas to help readers understand their mathematical principles. For some assumptions, clarify their rationality and basis.

2. Elaborate on the selection method and basis of the regularization parameter in SVF and the balance coefficient in WLSO. It is feasible to compare the effects of different parameter values on the fusion results through experiments and provide recommended parameter ranges.

3. The quantitative analysis indicators are not rich enough. We hope to add some evaluation indicators such as the message fusion quantity QAB/F. It should be noted that adding different evaluation indicators may lead to optimal conflicts, and we can truthfully explain the reasons.

4. The research motivation of the paper indicates that this method addresses the existing complex computational difficulties. However, the algorithm complexity analysis was not provided in the experimental section of this paper. (The running time can indirectly substitute for the complexity analysis.)

5. The references in the introduction section of the paper are too outdated. We hope to supplement or replace some with the latest literature, such as FusionOC, FusionPID, FusionCPP, etc. The more, the better.

Reviewer #4: This paper presents an infrared and visible image fusion method based on sub-window variance filter and weighted least squares optimization.

The studied topic is meaningful, and the proposed method seems reasonable.

The authors may further improve the paper from the following aspects.

Some overview papers are suggested to be given for better referring of the relevant topics, for example visual quality and its modeling, e.g., ‘Perceptual image quality assessment: a survey’; ‘Perceptual video quality assessment: a survey’; ‘Screen content quality assessment: overview, benchmark, and beyond’.

Multimodal fusion as well as visual saliency are introduced in this paper, which have been involved in many studies (e.g., ‘Study of subjective and objective quality assessment of audio-visual signals’; ‘Fixation prediction through multimodal analysis’; ‘A multimodal saliency model for videos with high audio-visual correspondence’). The authors may give some discussions on this aspect and the related works.

In the literatures, besides quality metrics designed for image fusion quality assessment, there are also many general quality visual quality metrics, e.g., BPRI (Blind quality assessment based on pseudo-reference image), BMPRI (Blind image quality estimation via distortion aggravation), RichIQA (Exploring rich subjective quality information for image quality assessment in the wild), which are also suggested to reviewed and discussed.

**Do you want your identity to be public for this peer review?** For information about this choice, including consent withdrawal, please see our Privacy Policy

Reviewer #1: No

Reviewer #2: No

Reviewer #3: **Yes: ** Linlu Dong

Reviewer #4: No

---

## [Author Response · Author response to Decision Letter 1]

17 Mar 2025

I'm truly grateful for the opportunity to revise my manuscript submitted to PLOS ONE. I have carefully reviewed all the comments from reviewers and the editor, and have uploaded a detailed response document to the submission system. This document comprehensively addresses each comment, along with explanations of the corresponding revisions made in the manuscript.

I firmly believe that with these modifications, the revised manuscript has significantly improved in terms of quality and clarity. I'm looking forward to your further evaluation and hope that it can meet the high standards of PLOS ONE.

---

## [Decision Letter · Decision Letter 1]

Saliency-Enhanced Infrared and Visible Image Fusion via Sub-Window Variance Filter and Weighted Least Squares Optimization

PONE-D-24-56535R1

Dear Dr. Li,

We’re pleased to inform you that your manuscript has been judged scientifically suitable for publication and will be formally accepted for publication once it meets all outstanding technical requirements.

Kind regards,

Mahmoud Emam, Ph.D.

Academic Editor

PLOS ONE

Additional Editor Comments (optional):

**Comments from PLOS Editorial Office** : We note that one or more reviewers has recommended that you cite specific previously published works in the current and previous rounds of revision. As always, we recommend that you please review and evaluate the requested works to determine whether they are relevant and should be cited. It is not a requirement to cite these works and you may remove any added citations before the manuscript proceeds to publication. We appreciate your attention to this request.

Reviewers' comments:

Reviewer's Responses to Questions

**Comments to the Author**

Reviewer #1: (No Response)

Reviewer #2: (No Response)

Reviewer #3: All comments have been addressed

2. Is the manuscript technically sound, and do the data support the conclusions?

Reviewer #1: (No Response)

Reviewer #2: (No Response)

Reviewer #3: Yes

3. Has the statistical analysis been performed appropriately and rigorously?

Reviewer #1: (No Response)

Reviewer #2: (No Response)

Reviewer #3: Yes

4. Have the authors made all data underlying the findings in their manuscript fully available?

Reviewer #1: (No Response)

Reviewer #2: (No Response)

Reviewer #3: Yes

5. Is the manuscript presented in an intelligible fashion and written in standard English?

Reviewer #1: (No Response)

Reviewer #2: (No Response)

Reviewer #3: Yes

Reviewer #1: The article titled “Saliency-Enhanced Infrared and Visible Image Fusion via Sub-Window Variance Filter and Weighted Least Squares Optimization”.

This revision has significantly improved the manuscript, and the authors have satisfactorily addressed my concerns. However, regarding the highlighted additions, I did not see the points made in recommendation number 3, which were marked in blue. Nonetheless, these recent changes have captured my attention.

Reviewer #2: authors addressed all comments

authors addressed all comments

authors addressed all comments

authors addressed all comments

Reviewer #3: (No Response)

**Do you want your identity to be public for this peer review?** For information about this choice, including consent withdrawal, please see our Privacy Policy

Reviewer #1: No

Reviewer #2: No

Reviewer #3: **Yes: ** Linlu Dong

---

## [Editor Report · Acceptance letter]

PONE-D-24-56535R1

PLOS ONE

Dear Dr. Li,

I'm pleased to inform you that your manuscript has been deemed suitable for publication in PLOS ONE. Congratulations! Your manuscript is now being handed over to our production team.

Kind regards,

on behalf of

Dr. Mahmoud Emam

Academic Editor

PLOS ONE